# Immune-Checkpoint Blockade Therapy in Lymphoma

**DOI:** 10.3390/ijms21155456

**Published:** 2020-07-30

**Authors:** Ayumi Kuzume, SungGi Chi, Nobuhiko Yamauchi, Yosuke Minami

**Affiliations:** 1Department of Hematology, National Cancer Center Hospital East, Kashiwa 277–8577, Japan; akuzume@east.ncc.go.jp (A.K.); schi@east.ncc.go.jp (S.C.); noyamauc@east.ncc.go.jp (N.Y.); 2Department of Hematology, Kameda Medical Center, Kamogawa 296–8602, Japan

**Keywords:** hematologic malignancies, immunotherapy, programmed cell-death protein 1 (PD-1)

## Abstract

Tumor cells use immune-checkpoint pathways to evade the host immune system and suppress immune cell function. These cells express programmed cell-death protein 1 ligand 1 (PD-L1)/PD-L2, which bind to the programmed cell-death protein 1 (PD-1) present on cytotoxic T cells, trigger inhibitory signaling, and reduce cytotoxicity and T-cell exhaustion. Immune-checkpoint blockade can inhibit this signal and may serve as an effective therapeutic strategy in patients with solid tumors. Several trials have been conducted on immune-checkpoint inhibitor therapy in patients with malignant lymphoma and their efficacy has been reported. For example, in Hodgkin lymphoma, immune-checkpoint blockade has resulted in response rates of 65% to 75%. However, in non-Hodgkin lymphoma, the response rate to immune-checkpoint blockade was lower. In this review, we evaluate the biology of immune-checkpoint inhibition and the current data on its efficacy in malignant lymphoma, and identify the cases in which the treatment was more effective.

## 1. Introduction

With the advent of research on the immune system, immune escape has been found to play a critical role in tumor survival. Tumor cells can evade the immune response for survival using programmed cell-death protein 1 (PD-1)-programmed cell-death protein 1 ligand 1 (PD-L1) immune checkpoint. T cells require two signals to perform effector functions. “Signal 1” is induced upon interaction between the T-cell receptor (TCR) and a major histocompatibility complex (MHC)-bound antigen presented on the surface of professional antigen-presenting cells (APCs). “Signal 2” is a co-stimulatory signal controlled by the binding of B7-1 (CD80) or B7-2 (CD86) on the surface of the APCs to CD28 present on the surface of the T cells, and in the absence of the second stimulus, the T cells become anergic [1]. Additionally, recent studies indicated that inflammatory cytokines including IL-12 or type 1 IFN provide a critical signal to enhance T-cell clonal expansion (“Signal 3”). CD8^+^ cytotoxic T lymphocytes (CTLs) survey the antigen in peripheral tissues, recognize the antigens presented by MHC class I, and perform target cell lysis. This immune function is regulated by central and peripheral checkpoints. Cytotoxic T-lymphocyte-associated antigen 4 (CTLA-4) is expressed on the surface of naïve T cells and competes with CD28 for binding to B7-1 and/or B7-2, which induces inhibitory signaling, causes T-cell exhaustion, and reduces cytotoxicity. In the tumor environment, activated CD8^+^ cytotoxic T cells recognize the target antigen peptide on tumor cells and initiate tumor-cell lysis, while tumor cells express PD-L1 and/or PD-L2, which bind to PD-1 on T cells and induce inhibitory checkpoint signaling. Various pathogens and tumors express these inhibitory checkpoints, thereby suppressing immunogenicity and minimizing detection. Based on reports, two additional immune-checkpoint molecules are present: T cell immunoglobulin and mucin domain-containing protein-3 (TIM-3) and lymphocyte activation gene-3 (LAG-3) [2]. TIM-3 is a type I transmembrane protein expressed on several types of immune cells, particularly CD4^+^ Th1 and CD8^+^ cytotoxic T cells, and has the ability to limit the duration and magnitude of T cell responses. TIM-3 expression is often associated with T-cell dysfunction and poor prognosis in certain tumor types. TIM-3 expression has been detected in hematologic malignancies such as adult T-cell leukemia/lymphoma and extranodal natural killer (NK)/T-cell lymphoma [3]. LAG-3 is a member of the immunoglobulin superfamily and acts as a negative regulator of T-cell homeostasis; LAG-3 was originally observed to be expressed in activated CD4^+^, CD8^+^, and NK cell subsets [4]. LAG-3 binds to MHC class II with a higher affinity than that of CD4, and LAG-3 expressed in cytotoxic T cells and NK cells binds to LSECtin, which is commonly expressed in various tumors. LAG-3 is expressed in TILs of several tumor types, such as breast, ovarian, and lung cancers, and its expression is related to the presence of PD-1^+^ T cells [5]. LAG-3 has been shown to be expressed in intratumoral T cells in hematologic malignancies such as Hodgkin lymphoma (HL), particularly in malignant Reed-Sternberg (RS) cell-rich sites [6]. In follicular lymphoma, a subset of intratumoral PD-1^+^ T cells was also found to be LAG-3^+^, and its presence was associated with that of functionally exhausted T cells. Clinically, LAG-3 expression on intratumoral T cells correlated with a poor outcome in FL patients [7]. In diffuse large B-cell lymphoma (DLBCL), LAG-3 was detected by the positive staining in 39% of tumor cells, while TIM-3 expression was associated with poor prognosis. Co-culture of DLBCL cell lines with primed T cells in the presence of anti-LAG-3 and anti-TIM-3 induced potent dose-dependent increments in in vitro cell death, suggesting the anti-tumor activity of these antibodies [8].

CTLA-4 and PD-1 blockade is a reasonable strategy for cancer treatment. Patients with metastatic melanoma received immune-checkpoint blockade therapy with agents such as ipilimumab and nivolumab, and objected response and prolonged survival were observed [9,10,11,12]. The response rate in melanoma patients treated through the PD-1 blockade approach was 40%; however, the response rate to CTLA-4 blockade was ~10%. A higher frequency of toxicity was observed in cases of CTLA-4 blockade than in those of PD-1 blockade. Anti-PD-1 antibodies inhibit the interaction between PD-1 and PD-L1/PD-L2, whereas anti-PD-L1 antibodies only inhibit the interaction between PD-1 and PD-L1. Although PD-L2 expression is generally restricted, it is known to increase in certain types of tumors (HL and primary mediastinal B-cell lymphoma).

Immune-checkpoint blockade therapy can be used to treat a wide range of tumors, such as non-small cell lung cancers and various hematological malignancies. The Food and Drug Administration has approved the use of anti-PD-1 (nivolumab and pembrolizumab), anti-PD-L1 (atezolizumab, avelumab, and durvalumab), and anti-CTLA-4 (ipilimumab and tremelimumab) agents.

Other immunotherapeutic strategies have also been studied in recent years, most notably chimeric antigen receptor (CAR) T-cell therapy and bispecific T-cell engager (BiTE) agents.

Although Immune-checkpoint blockade therapies have transformed the treatment landscape for patients with many advanced malignancies, the advantageous clinical outcomes associated with immune-checkpoint blockade therapies can be offset by potentially severe immune-related adverse events (irAE). Because irAE are often distinctly different from the classical chemotherapy-related toxicities, clinicians have to be aware of the clinical presentation, diagnosis, and management of these toxicities.

The results of several interesting immunotherapies have been reported. Here, we review the investigations of the immune-checkpoint strategy. 

## 2. HL 

RS cells, the signature cells in HL, are scattered in a background of inflammatory cells that contain an abundance of CD8^+^ CTLs and CD4^+^ T-cells including immunosuppressive regulatory T cells (T_reg_) [13,14]. RS cells overexpress PD-L1 and PD-L2, while tumor-infiltrating lymphocytes (TILs) express PD-1. PD-L1 and -L2 bind to PD-1 and induce immunosuppression in the tumor microenvironment [15]; subsequently, CD8^+^ CTLs proliferation is limited. RS cells also secrete several chemokines that attract Tregs by which RS cells escape elimination by CD8^+^ CTLs [16]. The role of CD4^+^ T-cells other than T_reg_ (T helper cells) on immune evasion in HL has not been fully elucidated [17]; however, inhibitory cytokines, such as transforming growth factor beta (TGFβ) secreted by RS cells, have a distinct suppressive impact on T helper cells [18]. PD-1 checkpoint blockade can contribute to the restoration of TIL functions. Therefore, blockade of inhibitory signals originating from PD-1 ligands is an attractive strategy in HL.

The chromosomal region 9p24.1 contains *PDL1* and *PDL2*. In nodular sclerosing HL, this region is amplified, which induces the expression of *PDL1* and *PDL2* in biopsy specimens. The amplification of 9p24.1 is associated with shorter progression-free survival (PFS) [19]. In addition, the 9p24.1 amplicon includes *JAK2*. The JAK/STAT signal cascade induces the subsequent transcription of *PDL1* [20]. 

In Appendix A, we show results from clinical trials of PD-1 blockade in cHL. The efficacy of the anti-PD-1 antibodies, pembrolizumab, and nivolumab, in HL has been reported previously.

In CheckMate-039, a phase I study, the efficacy and favorable tolerability of nivolumab and pembrolizumab were demonstrated in patients with HL. Twenty-three patients with relapsed and/or refractory HL, including 18 patients who had received autologous stem cell transplant (ASCT) and brentuximab vedotin (BV) previously, were administered nivolumab at a dose of 3 mg/kg every 2 weeks until disease progression or complete response was observed, or for a maximum of 2 years. The overall response rate (ORR) was 87% (complete response (CR) = 17%), and 6-month PFS was 86%. After a median follow-up duration of 40 weeks, the median overall survival (OS) could not be achieved. The 10 patients with tumor samples available had *PDL1* and *PDL2* amplifications, as revealed by fluorescence in situ hybridization experiments, and the levels of PD-L1 and PD-L2 were increased, as detected using immunohistochemical analysis [21]. In CheckMate-205, a phase II study, 80 HL patients with prior failure in ASCT and BV therapy received 3 mg/kg of nivolumab intravenously every 2 weeks. The ORR was 66.3% (CR = 7%) at a median follow-up duration of 8.9 months [22]. In a phase II study (KEYNOTE-087), the efficacy of pembrolizumab (200 mg administered intravenously every 3 weeks) was demonstrated in 210 patients with r/r HL. The ORR was 69% (CR = 22%), and the PFS was 72% at 6 months [23,24]. The correlation studies revealed that PD-L1 or PD-L2 were overexpressed in the RS cells in all available tissue samples. Based on the notable success of anti-PD-1 antibody monotherapy in relapse/refractory (r/r) HL patients, it can be used as a frontline therapeutic agent for HL. 

The efficacy of combination therapy is expected. In CheckMate-205, a phase II study, 51 patients (cohort D) received AVD chemotherapy (doxorubicin, vinblastine, and dacarbazine) for six cycles and nivolumab (240 mg, administered intravenously, every 2 weeks for four cycles). The ORR was 84% (CR = 67%) and the 9-month PFS was 92%. In this study, it was reported that patients with higher-level RS PD-L1 expression elicited more favorable responses [25]. German Hodgkin Study Group reported the result of a phase 2 study in which nivolumab plus AVD showed the high CR rate of 92% for patients with early-stage unfavorable HL [26]. In a phase 2 study of combination treatment with BV and nivolumab in 21 treatment-naïve HL patients who were above 60 years of age, patients received 1.8 mg/kg of BV and 3 mg/kg of nivolumab intravenously every 3 weeks for up to 16 cycles. The ORR was 100%, and 72% achieved CR. In this study, one of the patients developed *Pneumocystis jirovecii* pneumonia, and one developed grade 3 acute renal failure and sepsis. The efficacy of nivolumab plus BV was also reported for patients with r/r HL. In phase 1/2 studies, 62 patients with r/r HL received BV (1.8 mg/kg IV) and nivolumab (3.0 mg/kg IV) for four cycles. The ORR was 82% (CR = 61%) [27]. 

The efficacy of concomitant or sequential treatment with immune-checkpoint inhibitor and local radiation therapy was also reported. Quéro showed the case series of four patients with r/r HL who received immune-checkpoint inhibitor and consolidative radiotherapy to mediastinal disease. At the median follow-up of 13 months, all four patients were alive with CR. Importantly, these four patients experienced grade 1–2 lung toxicities. Further study with a larger patient cohort is warranted to define the optimal timing of radiotherapy [28]. Another case report showed a clinical course of a patient with r/r HL who experienced complete local and abscopal responses from nivolumab plus radiation therapy [29]. Combining radiotherapy with immune-checkpoint inhibitors might be a cost-effective strategy, allowing for earlier stopping of immunotherapy in patients in complete remission, at least for patients with localized r/r HL. 

According to the aforementioned two largest prospective studies that evaluated nivolumab (CheckMate-205) and pembrolizumab (KEYNOTE-087) in R/R HL, the overall response rate and complete response (CR) rate were around 70% and 20%. Although the long-term follow-up data are still lacking, a fraction of patients in CR seems to experience durable remissions, which raises question about the need for continuing therapy. Manson retrospectively analyzed 87 r/r HL patients treated with nivolumab, of which 11 patients discontinued nivolumab (*n* = 7, due to prolonged remission; *n* = 4, due to toxicities) [30]. The median duration of exposure to nivolumab was 13.8 months in 7 patients who had prolonged remission. At the median follow-up of 21.2 months (from nivolumab discontinuation), six out of seven patients were alive in CR, and the remaining one patient achieved PR upon nivolumab retreatment. Further studies are warranted to elucidate biomarkers which enable the capture of a fraction of patients who may be in remission after discontinuation of the immune-checkpoint inhibitor. Defining these biomarkers is particularly important because the remission rates are high and most of these patients are relatively young. 

Clinical studies mentioned above showed the rationale of combining the immune-checkpoint inhibitor with chemotherapy or radiotherapy which, at the same time, may reduce the risk of long-lasting toxicities. Further studies are needed to extrapolate the efficacy and the safety of these combination therapies for hematologic malignancies other than HL.

## 3. DLBCL

The tumor cells in DLBCL rarely express PD-L1, as observed in flow cytometry experiments (3 of 28 cell lines) [31]. Andorsky also reported that 19 were of the GCB type, only one of which expressed PD-L1, whereas 8 of the 14 ABC types expressed PD-L1. Menter reported that 80 of 260 (31%) cases had PD-L1 expression, which correlated with the number of PD-1 positive T cells of the ABC type and inversely correlated with the number of FOXp3-positive T_reg_ cells of the GCB type [32]. Kiyasu reported that PD-L1 expression was associated with both ABC-DLBCL types (*p* < 0.0001). TIM-3 expression also increased in peripheral blood CD3^+^ T cells in DLBCL patients and was associated with DLBCL staging [33].

In Appendix A, we show results from clinical trials of PD-1 blockade in DLBCL. In a phase I trial of nivolumab, it was administered to 11 patients with r/r DLBCL. The ORR was 36% (CR = 18%), and the median PFS was 7 weeks [34]. In a phase II trial, 66 patients received pidilizumab (1.5 mg/kg every 42 days) after ASCT; the CR rate was 34%, and ORR was 51% [35].

In a phase II study, patients with r/r DLBCL who were ineligible for ASCT or in whom ASCT was ineffective received 3 mg/kg nivolumab every 2 weeks. The ORR was 10% in the ASCT-failed cohort and 3% in the ASCT-ineligible cohort. The median PFS was 1.9 and 12.2 months, and OS was 1.4 and 5.8 months, respectively. Of the evaluable samples in the analysis of 9p24.1, 16% exhibited low-level copy gain and 3% contained amplifications. Since genetic alterations in 9p24.1 are infrequent, the response rate of DLBCL might have been low [36].

In a phase I study, ipilimumab was administered to patients with r/r B-cell lymphoma at 3 mg/kg and then at 1 mg/kg per month for 3 months, with subsequent escalation to 3 mg/kg per month for 4 months. Eighteen patients were treated, and one patient with DLBCL achieved CR [37]. 

Recent reports suggest that combination treatment, such as with chimeric antigen receptor T cells (CAR-T) and nivolumab, and PD-1 inhibition with concomitant radiotherapy [38].

Although biomarkers are necessary to assess the efficacy and predict prognosis, these are absent in DLBCL patients undergoing immunotherapy. However, the plausible association between PD-L1 expression and DLBCL prognosis remains inconclusive and controversial [39].

## 4. Other Large B-Cell Lymphomas

Primary central nervous system lymphoma (PCNSL) and primary testicular lymphoma (PTL) are extranodal large B-cell lymphomas that respond poorly to contemporary treatment regimens. Bjoem reported that PCNSL and PTL frequently exhibit 9p24.1/PD-L1/PD-L2 copy number alterations and translocations [40]. They evaluated 43 PTL, 43 EBV-negative PCNSL, and 8 EBV-positive PCNSL samples, and observed that >50% of the PCNSL and PTL samples harbored alterations in chromosome 9p24.1.

Primary mediastinal large B-cell lymphoma (PMBCL) is characterized by a type 2 T-helper cell (T_H_2)-skewed cytokine profile and constitutive activation of nuclear factor κB (NF-κB), and this environment may induce the inhibition of CTLs via the PD-1-PD-L1 axis. Amplification of the 9p24.1 locus was observed in 63% of PMBCL cases [20,41], and chromosomal rearrangements involving 9p24.1 were detected in 20% PMBCL samples [42,43]. The presence of genetic alterations enhances the expression of PD-L1 and PD-L2. Other structural rearrangements involving JAK2, CIITA, and REL were detected in PMBCL patients [44,45]. 

In a phase I study on pembrolizumab in r/r PMBCL patients (Keynote-013), 21 patients received pembrolizumab, and the ORR was 48% (CR was 33%). The median PFS was 10.4 months and OS was 31.4 months. In Keynote-170, a phase II trial, r/r PMBCL patients received pembrolizumab; the median PFS was 5.5 months and the median OS was not achieved [46]. Among 42 evaluable samples, the magnitude of abnormality in the 9p24 region was associated with PD-L1 expression, which in turn was significantly associated with PFS. The findings from these trials indicate the safety of the PD-1 blockade and its anti-tumor activity in r/r PMBCL. However, the number of cases is limited and studies on a larger cohort are awaited.

## 5. Follicular Lymphoma 

The tumor microenviroment in follicular lymphoma contains CD8^+^ CTLs and T_reg_ cells [47]. It was reported that the presence of PD-1 positive T cells and CD14^+^ follicular dendritic cells are related to the duration of transformation in patients with follicular lymphoma [48]. PD-1 is expressed in the germinal centers of follicular cells; however, expressions of PD-1 and PD-L1 on follicular lymphoma tumor cells are rare. TILs increase the expression of PD-1 and inhibit cytokine signaling compared to the action of peripheral blood T cells [49]. In patients with solid tumors, the significantly PD-1 positive TILs are associated with advanced stages and poor prognosis; however, the number of PD-1 positive TILs in follicular lymphoma is associated with favorable outcomes, independent of follicular lymphoma International Prognostic Index risk stratification [50]. Carreas also reported that patients with PD-1 positive cells ≦5% were at a higher risk of histologic transformation, and a lower percentage of PD-1 positive cells were present in transformed DLBCL than in follicular lymphomas.

However, Richendolllar reported that the increase in the number of PD-1 positive T_FH_-cells is an independent, poor prognostic risk factor in follicular lymphoma [51]. In the tumor microenvironment in follicular lymphoma, several types of T cells, such as CD4^+^ T_H_1 cells, CD8^+^ CTLs, and T_reg_ cells are present; hence, the apparent inconsistencies between the results might reflect the fact that immunohistochemical analysis was used to assess PD-1 expression, which consequently prevents the accurate determination of the relative T-cell subsets that are PD-1 positive. PD-1 blockade might enhance the effects of anti-tumor CD8^+^ CTLs; however, the effects of other PD-1 positive cells are unclear. Therefore, the efficacy of PD-1-PD-L1 axis blockade therapy was investigated in clinical trials.

In Appendix A, we show results from clinical trials of PD-1 blockade in FL. In a phase I study, patients with r/r follicular lymphoma received nivolumab (1 or 3 mg/kg, every 2 weeks) [52]. The ORR was 40%, CR was 10%, and the 24-week PFS rate was 68%. In a phase II study, 32 patients with r/r follicular lymphoma underwent combination therapy with pidilizumab and rituximab. The ORR of evaluable patients was 66% and the CR was 52%. The median PFS was 18.8 months, and no autoimmune or treatment-related adverse events of grade ≧3 occurred. Westin et al. hypothesized that the lack of immune-related adverse events could be attributed to the low-frequency dosage of pidilizumab compared to that of other PD-1 blocking antibodies, B-cell depletion by rituximab, and the immunocompromised state of patients with follicular lymphoma. In this study, the expression of PD-L1 in peripheral blood T cells was significantly higher in responders than in non-responders, although it was not associated with PFS [52].

## 6. Epstein-Barr Virus (EBV)-Associated Lymphoma 

EBV is one of the most common human viruses. EBV has been implicated in the development of a wide range of cancers, such as Burkitt lymphoma, T-cell and NK cell proliferation, and a subset of DLBCL [53]. EBV-encoded proteins function as various cellular factors associated with cell growth, transcription, and apoptosis, and regulate diverse homeostatic cellular functions.

Viral infection has been observed to induce PD-L1 expression, such as in human T-lymphotropic virus-1 (HTLV-1)-associated adult T-cell leukemia/lymphoma [54]. In classic HL, the EBV-latent membrane protein (LMP1) induces PD-L1 expression via the AP-1 and JAK-STAT pathways [55]. In DLBCL tumors, PD-L1 was found to be expressed in 65–100% cases [56,57]. A high frequency of PD-L1/PD-L2-associated genetic aberrations was observed in EBV-positive lymphoma patients (22%) [58]. Additionally, EBV-positive DLBCL is reportedly characterized by frequent mutations in TET2 and DNMT3A and the paucity of alterations in CD79B, MYD88, CDKN2A, and FAS.

*EBV* gene products activate the NF-κB pathway and hinder the p16 (encoded by CDKN2A)-Rb pathway [59]. Mutations in TET2 and DNMT3A indicate the involvement of deregulated DNA methylation and demethylation processes in EBV-positive DLBCL [60]. Tosagaratau reported that TET2 disruption resulted in aberrant proliferation depending on the antigen receptor [61]. These mutations may contribute to tumor growth, and the relationships between PD-L1/PD-L2 and FAS alterations suggest the varying roles played by these molecules in the escape from T-cell-mediated immune reactions against virally infected cells.

In Appendix A, we show results from clinical trials of PD-1 blockade in Virus-associated lymphoma and other types of lymphoma. Seven patients with r/r NK/T cell lymphoma received pembrolizumab and five patients achieved CR [62]. In NK/T cell lymphoma, PD-L1 is reported to be upregulated by EBV-driven LMP1 through the NF-κB pathway and PD-L1 expression correlated with poor prognosis [63]. In another study, 30 patients with r/r NHL received pembrolizumab, and 7 patients with EBV-positive NHL responded to the treatment. Furthermore, PD-L1 expression was higher in EBV-positive (56%) than in EBV-negative NHL (11%, *p* < 0.001) [64]. Therefore, immune-checkpoint blockade may serve as an attractive choice of therapy in EBV-positive lymphoma.

## 7. HIV-Associated Lymphoma 

Acquired immunodeficiency syndrome is commonly associated with malignancies, such as cervical carcinomas, Kaposi sarcoma, lymphomas, DLBCL, primary effusion lymphoma, Burkitt lymphoma, PCNSL, and HL. EBV and/or HHV8 co-infection is observed in several HIV-positive patients [65]. The prognosis in HIV-associated lymphoma has been poor compared to that in non-HIV-associated lymphoma.

The function of CD8^+^ T cells is impaired in HIV-positive patients, and the cytokine secretion capacity is reduced. PD-1 is upregulated in HIV-specific CD8^+^ T cells and the phenomenon is correlated with the HIV viral load. PD-1 engagement blockade using PD-L1 enhanced the survivability and proliferative capacity of HIV-specific CD8^+^ T cells and enhanced the production of cytokines and cytotoxic molecules in response to cognate antigen recognition. As a result of chemotherapy, the risk of infection is higher in patients with HIV-positive lymphoma [66]. PD-L1 expression is higher in B cells in HIV-positive patients [67]. HIV infection alone or with EBV/HHV8 co-infection is associated with PD-L1 expression in neoplastic B cells, notably in post-germinal-center lymphomas, such as ABC-DLBCL, primary effusion lymphoma, and plasmablastic lymphoma [68]. The CD4^+^ T cell count reduces in an HIV-infected patient and the patient becomes immunosuppressed, which gradually worsens with chemotherapy. Antiretroviral agents have overlapping toxicities and drug interactions with chemotherapeutic agents, and the toxicity profile of anti-PD-1 and anti-PD-L1 antibodies is not associated with further immunosuppression. Therefore, the immune-checkpoint blockade is one of the attractive therapies for HIV-associated lymphoma. Chang reported that two patients with HL and HIV infection who received nivolumab could achieve CR [69]. The efficacy and safety need to be evaluated in additional studies.

## 8. After Autollogous Transplantation

A total of 66 Patients with DLBCL after auto-HCT received PD-1 antibody pidilizumab as maintenance, and PFS was 0.72 after 16 months. Among the 35 patients with measurable disease after auto-HCT, the ORR after pitilizumab was 51%. The toxicity was mild, and circulating lymphocyte subsets were increased. It may be associated with a promising therapeutic strategy [35].

## 9. Before/After Allogenic Transplantation

An international retrospective analysis reported 39 patients with lymphoma received prior treatment with a PD-1 inhibitor before allogenic hematopoitetic stem cell transplant. The median time before HSCT was 62 days (7–260). The 1-year cumulative incidence of grade3–4 graft-versus-host disease (GVHD) was 23%, and there were 4 treatment-related deaths. Despite early toxicites, 1 year OS and PFS rates were 89% and 76%, respectively [70].

Several papers have reported that the use of immune-checkpoint inhibitors after allo-HSCT increases GVHD. In murine models, PD-1 blockade led to an increase in CD4^+^ and CD8^+^ T-cell immune responses and enhanced GVHD via an IFN-γ dependent mechanism [71]. CTLA-4 blockade enhances the graft-versus-tumor (GVT) effect without worsening GVHD [72]. It is also reported that blockade of PD-1/PD-L1 is more likely to induce GVHD than PD-1/PD-L2 [73]. PD-L1/CD80 interaction enhances the proliferation of reactive T cells and induces GVHD [74].

With low-dose ipilimumab (0.1–3.0 mg/kg) in patients with relapse after allo-HSCT in the phase 1 study, there was no clinically significant induction of GVHD, and three patients with lymphoid malignancies achieved responses [75]. Subsequent phase 1/1b studies have allowed patients above the stable stage to receive ipilimumab (3–10 mg/kg) 4 times every 3 weeks, and patients with stable disease or better could receive maintenance doses every 3 months for a year [76]. Also, 5/28 patients (23%) had a complete response. Responses were associated with in situ infiltration of cytotoxic CD8^+^ T cells, decreased activation of Treg, and the expansion of subpopulations of effector T cells. In this trial, GVHD responsive to corticosteroids developed in 4 of 28 patients. The 1-year OS rate was 49%.

In another study, 20 HL patients relapsing after allo-HSCT received nivolumab (3 mg/kg, every 2 weeks). Six patients (30%) had nivolumab-induced GVHD, and two patients died as a result of GVHD. The 1-year PFS was 58.2% and the OS rate was 78.7% [77]. The authors concluded nivolumab is safe to use after allo-HSCT. In contrast, another US retrospective analysis of 31 lymphoma patients relapsed after allo-HSCT receiving PD-1 blockade revealed a high frequency of GVHD (17/31 patients) and 8 patients died due to GVHD [78].

The number of cases is still low, and there are no recommendations for the use of immune checkpoint inhibitors before and after allo-HSCT, the optimal drug, dose, or timing of administration. Therefore, it would be difficult to use immune-checkpoint inhibitors before/after allo-HSCT outside of the context of a clinical trial.

## 10. Other Immune Therapies

CAR-T-cell therapy is a form of personalized cancer treatment. CARs are formed by combining the antigen-binding site of an antibody with the intracellular domain of a T cell activation receptor using genetic recombination technology. The CAR gene is introduced into the T cell genome through gene transfer technology. When the CAR-T cells recognize the antigen in the target cell, the intracellular domain of TCR cells is directly stimulated, which induces a specific and powerful immune response against the target antigen. CAR-T cells do not require the intervention of HLA molecules for antigen recognition, which makes them more effective against tumor cells that have lost HLA expression owing to the tumor immune evasion mechanism. Irving and Weiss reported that CAR composed of a CD8 and CD3ζ chain could mediate T-cell activation independent of endogenous TCR [79].

The first generation CAR comprises a single-chain antibody (scFV) consisting of a light chain (VL) and a heavy chain in the variable monoclonal antibody against tumor-associated antigens in the extracellular region, a transmembrane domain in the transmembrane region, and a T cell receptor ζ chain in the endodomain of the intracellular region as antigen-binding sites. The first generation CAR did not exert distinct anti-tumor effects in clinical trials on renal cell carcinoma, neuroblastoma, and ovarian cancer [80,81,82], owing to the lack of non-specific T cell activation signals (secondary signal) relayed via co-stimulatory molecules. The expression of co-stimulatory molecules is often downregulated in tumor cells, such that even if CAR-T cells are able to recognize the target antigen, T cell activation is insufficient. The second-generation CAR contains the secondary signal for T cell co-stimulatory molecules such as CD27, CD28, 4-1BB, and OX40. Third-generation CARs incorporated with two co-stimulatory molecules, such as CD28 and 4-1BB, or CD28 and OX40, are also being developed. In a trial on second-generation CAR-T cells, a patient with follicular lymphoma received CD28-containing CD19-CAR [83]. The tumor underwent partial regression, and the CD19-CAR transgene was detected in the peripheral blood sample for up to 27 weeks after infusion. In a trial on 4-1BB-containing CD19-CAR, patients with relapsed B-cell CLL were administered autologous CAR-T cells after lymphodepleting chemotherapy [84,85]. The results were encouraging in spite of the high tumor burdens (among the three CLL patients treated, two patients achieved long term complete remissions and one patient achieved prolonged partial remission). In a phase 1-2a study, 30 patients with relapsed acute lymphoblastic leukemia (ALL) received autologous T cells transduced with CD19-directed CAR (CTL019) lentiviral vector. The six-month event-free survival rate was 67% and the OS was 78%. At 6 months, the probability that a patient would have persistence of CTL019 was 68% [86]. Maude also evaluated the efficacy in 75 patients who received an infusion of tisagenlecleucel. The rates of event-free survival and OS were 73% and 90%, respectively, at 6 months. Grade 3 or 4 adverse events suspected to be related tisagenlecleucel administration occurred in 73% of patients. The cytokine release syndrome was noted in 77% and 48%, respectively, of patients who received tocilizumab [87]. The efficacy of CAR-T therapy was reported in DLBCL patients [88,89]. Schuster reported that 28 patients with lymphoma received CTL019, among which 18 patients elicited a response (6 of 14 patients with DLBCL and 10 of 14 patients with FL). All patients who achieved complete remission within 6 months remained in remission for up to 37.9 months (median, 29.3 months) after induction. CAR-T therapy can be effective for treating hematologic malignancies such as CLL, ALL, and B-cell lymphoma. The efficacy of CAR-T therapy in AML has also been studied. However, unlike that for ALL, the target antigen is weakly expressed in AML cells and the target antigen is also expressed in normal myeloid cells in CAR-T therapy for AML. The development of safer and more effective CAR-T therapeutic strategies is currently underway. CAR-T cells have been developed against antigens such as Lewis Y, CD33, and NKG2D ligands; however, the efficacy was limited [90,91]. Research on CAR-T therapy for multiple myeloma is also underway. CAR-engineered T cells that target B-cell maturation antigen (BCMA), CD138, CS1 glycoprotein antigen (SLAMF7), and light chains are being developed actively for the treatment of r/r MM. In a phase I study, 24 patients with r/r MM were injected with BCMA-specific CAR-T cells (CT053) and the response rate was 87.5% (CR rate was 70.8%) [92]. CAR-T therapy will continue to serve as an attractive strategy for hematological malignancies.

Bispecific T-cell engager (BiTE) is a recombinant bispecific protein that has two linked scFVs from two different antibodies: one that targets antigens on the surface of malignant cells and the other that targets a cell surface molecule on T cells (such as CD3ε). BiTEs can bind tumor antigens to T cells simultaneously, and mediate T-cell responses and killing of tumor cells. The adherence of the target cell and T-cell was mediated by BiTE independent of the MHC haplotype. In BiTE treatment, antigen-experienced T-cell subsets mediate the BiTE-induced tumor cell death; however, naïve T cells are not activated [93]. It has been demonstrated that BiTEs can induce the sequential killing of tumor cells by T cells, where one T cell can kill several tumor cells. BiTE-redirected function is dependent on the dose of BiTE administered. BiTEs also can mediate the secretion of cytokines that induce changes in the tumor microenvironment and endogenous anti-tumor immunity [94]. Substantial numbers of redirected T cells can be provided by a large pool of antigen-experienced T cells. After BiTE administration, the number of circulating T cells is observed to increase, which indicates the expansion of the pool of T cells that act against tumor cells [95,96].

Blinatumomab was first used clinically for the treatment of CD19^+^ Philadelphia chromosome-negative (Ph^-^) r/r B-ALL. A total of 189 patients were treated with blinatumomab, and 81 patients (43%) achieved CR/CRh after two treatment cycles. Thirty-two (40%) of the patients who achieved CR/CRh underwent allogeneic HSCT [97]. In a phase II trial (ALCANTARA), 45 patients with Ph^+^ r/r B-ALL received blinatumomab. Sixteen patients (36%) achieved CR; the median duration of remission was 6.7 months and OS was 7.1 months. In another phase III trial (TOWER), blinatumomab treatment significantly improved prognosis and led to higher rates of hematologic remission in patients with Ph^-^ r/r B-ALL than observed in the chemotherapy group [98]. The median OS was 7.7 and 4.0 months in the chemotherapy group. Remission rates in 12 weeks following treatment initiation were higher in the blinatumomab group than in the chemotherapy group (CR with full hematologic recovery rate was 34% vs. 16%, *p* < 0.001).

In a phase I trial of blinatumomab in patients with r/r NHL [99], blinatumomab monotherapy (60 µg/m^2^/day) yielded an OS of 69%. In a phase II trial, patients with r/r DLBCL received blinatumomab. Among 21 evaluable patients, the OS was 43% after one cycle; CR was 19%. In this trial, neurologic events such as tremors, encephalopathy, aphasia, speech disorder, dizziness, somnolence, and disorientation were observed as well [100].

## 11. Conclusions

We have summarized the current immunotherapies for hematologic malignancies. According to our findings, all the platforms have demonstrated efficacy in clinical trials against hematologic malignancies. This will improve our understanding of the anticancer mechanisms, treatment resistance, and combined treatment and chemotherapy, and it will also help improve the treatment outcomes in malignancies.

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
