# Peer review of "Immune-Checkpoint Blockade Therapy in Lymphoma"

_ijms, 2020, doi:10.3390/ijms21155456_

Round 1
Reviewer 1 Report
In this review, Kuzume et al provide a clear summary of the role of ICI in lymphoma. Overall, the text is clear and this review should be valuable for IJMS readers.
My suggestions to further improve the quality of the paper are;
- Tables summarizing past and ongoing clinical trials are required. It should be mentioned that which subtype respond well and which is not.
- In the Introduction, it may be better to refer to Signal 3. (otherwise, readers might be misled that T cells only require Signal 1 and 2)
- Line 89: In addition to CTLs and Treg, helper T cells (Th1 or cytotoxic CD4+ T cells) also infiltrated in HL?
- Is PD-L1 also high in NK/T cell lymphoma?
- Please discuss about when to discontinue the therapy in good responders, and the rationale of combining ICI with chemotherapy (and/or radiotherapy)
Author Response
Reviewer 1
Comments and Suggestions for Authors
In this review, Kuzume et al provide a clear summary of the role of ICI in lymphoma. Overall, the text is clear and this review should be valuable for IJMS readers.
My suggestions to further improve the quality of the paper are;
1. Tables summarizing past and ongoing clinical trials are required. It should be mentioned that which subtype respond well and which is not.
The authors appreciate the reviewer’s important suggestion. We add “Table 1” into the manuscript which summarizes past and ongoing clinical trials. We also add brief comments into column of Table 1 to enable readers understand which subtypes respond well and which are not.
2. In the Introduction, it may be better to refer to Signal 3. (otherwise, readers might be misled that T cells only require Signal 1 and 2)
The authors appreciate the reviewer’s important comment. We describe the role of signal 3 to avoid the readers’ misunderstanding. (line 30-31).
3. Line 89: In addition to CTLs and Treg, helper T cells (Th1 or cytotoxic CD4+ T cells) also infiltrated in HL?
Several studies showed that, besides CD8+ T-cells and Treg, CD4+ T-cells are major components of tumor-infiltrating lymphocytes in HL, while the role of CD4+ T-cell in immune evasion is not fully elucidated. We cited a study which showed CD4+ T-cell infiltration in HL tissue (line 93-94).
4. Is PD-L1 also high in NK/T cell lymphoma?
The authors appreciate the reviewer’s comment. Previous reports shows that PD-L1 is upregulated by EBV-driven LMP1 through NF-kB pathway. We described PD-L1 expression in NK/T-cell lymphoma (line 319-320).
5. Please discuss about when to discontinue the therapy in good responders, and the rationale of combining ICI with chemotherapy (and/or radiotherapy)
The authors appreciate the reviewer’s important suggestion. In hematological malignancy, there has been scarce evidence regarding the efficacy and safety of combination therapy consisting of immune-checkpoint inhibitor and chemotherapy/radiotherapy, except for HL.. So the authors propose that the discussion about the rationale of combining immune-checkpoint inhibitors with chemotherapy/radiotherapy could be added to “HL” section (line 142-216). Similarly, We also add the discussion about when to discontinue immune-checkpoint therapy in patients with CR into “HL” section.
Additionally, We add the discussion about immune-checkpoint inhibitor prior to or post stem cell transplantation (line 355-380).
Reviewer 2 Report
The authors presented the results of a number of clinical studies in which anti-PD-1 / PDL-1 antibodies were used in different hematological diseases.
In general the manuscript is well elaborated and written.
Despite spectacular successes in the treatment of some malignant neoplasms, immunotherapy also causes side effects leading even to the death of the patient.
The authors should present the disadvantages of immunotherapy in the Introduction.
I recommend the manuscript to be published after MINOR REVISION.
Author Response
Reviewer 2
Comments and Suggestions for Authors
The authors presented the results of a number of clinical studies in which anti-PD-1 / PDL-1 antibodies were used in different hematological diseases.
In general the manuscript is well elaborated and written.
Despite spectacular successes in the treatment of some malignant neoplasms, immunotherapy also causes side effects leading even to the death of the patient.
The authors should present the disadvantages of immunotherapy in the Introduction.
I recommend the manuscript to be published after MINOR REVISION.
The authors appreciate the reviewer’s important suggestion. We describe the disadvantage of immunotherapy (immune related AE) in the Introduction (line 77-82).